# Measurement Invariance of the Multidimensional Jealousy Scale and Quality of Relationships Inventory (Friend)

**DOI:** 10.3390/bs14010044

**Published:** 2024-01-10

**Authors:** Ãngela Leite, Beatriz Silva, Beatriz Vilela, Inês Rodrigues, Joana Fernandes, Joana Romão, Ana Margarida Ribeiro

**Affiliations:** 1Centre for Philosophical and Humanistic Studies, Faculty of Philosophy and Social Sciences, Universidade Católica Portuguesa, Rua Camões, 4710-362 Braga, Portugal; anamargamr@gmail.com; 2Departamento de Educação e Psicologia, Escola de Ciências Humanas e Sociais, Universidade de Trás-os-Montes e Alto Douro, Quinta de Prados–Folhadela, 5000-801 Vila Real, Portugal; al68318@alunos.utad.pt (B.S.); beatriz.vilela@prochildcolab.pt (B.V.); al66575@alunos.utad.pt (I.R.); al69140@alunos.utad.pt (J.F.); al68266@alunos.utad.pt (J.R.)

**Keywords:** equivalence, factor, interpersonal quality, jealousy, validation

## Abstract

The aim of this study is to measure the invariance of the Multidimensional Jealousy Scale (MJS) and of the Quality of Relationships Inventory (Friend) (QRI-F) across gender, age, education, and being in a romantic relationship in a Portuguese sample (N = 662). A confirmatory factor analysis was performed to test the fit of different potential factor structures. The results pointed out that both MJS and QRI-F were most suitable if represented by three first-order factors correlated between them. Results from multi-group analyses suggested there was factorial invariance for these structures across groups, suggesting that the MJS and the QRI-F provide, respectively, an assessment of romantic jealousy and quality of relationship that are equivalent across gender, age, education, and being in a romantic relationship. The study established the strong psychometric properties of its instruments, validating reliability and convergent and discriminant validity, thereby bolstering the research’s overall credibility. Additionally, cognitive jealousy is primarily influenced by heightened conflict values, with education, relationship status, and gender moderating the associations between QRI-F dimensions and MJS behavioral and cognitive jealousy. The research offered in-depth perspectives on jealousy, underscoring its diverse manifestations across demographic variables and illuminating the complexities within the dynamics of friendships.

## 1. Introduction

Jealousy and the quality of relationships in friendships are interconnected aspects that can significantly influence the dynamics and overall health of social connections. The nature of jealousy in friendships can be manifested in various ways. It might arise from feelings of insecurity, fear of abandonment, or perceived threats to the relationship. It can be triggered by factors such as attention given to other friends, perceived favoritism, or changes in the friend’s life [1]. Jealousy impacts friendships; in fact, unaddressed jealousy can strain friendships, leading to communication breakdowns, mistrust, and even the deterioration of the relationship [2]. However, experiencing occasional jealousy is normal, and how individuals manage and communicate these feelings is crucial for maintaining healthy friendships [3]. In fact, while experiencing jealousy appears to be a common emotion in adolescence, with most teenagers feeling some level of jealousy at least once, adolescents who are prone to jealousy tend to express their feelings through relational aggression [4]. This typically happens when they perceive a third party as a threat to the quality or exclusivity of their friendship [5]. Open and honest communication is vital in addressing jealousy. Friends who can express their feelings, share concerns, and work together to find solutions are more likely to navigate jealousy in a constructive way [2]. Understanding each other’s perspectives can strengthen the bond.

Croucher et al. [6] discovered that individuals hailing from countries that prioritize self-centered thinking, a trait more prevalent in cultures with a masculine orientation, tend to display a greater tendency toward jealousy. This increased self-centered thinking stems from an inclination to focus more on one’s own well-being and less on the well-being of the couple [6]. Notably, both Ireland and the United States generally exhibit more masculine cultural characteristics compared to India and Thailand. Consequently, it is plausible that Americans and Irish individuals might manifest a higher degree of jealousy, given that jealousy is viewed as an emotional manifestation of competition [6]. Furthermore, Croucher et al. [7] conducted a comparative analysis of jealousy between India and the United States. Noteworthy variations emerged between men and women across all facets of jealousy. Specifically, Indians reported lower levels of both cognitive and emotional jealousy in comparison to Americans. Croucher et al. [7] also identified religion as a crucial factor in predicting jealousy. Hindus exhibited higher levels of both cognitive and emotional jealousy, while Christians demonstrated greater cognitive jealousy than Muslims.

While research often focuses on romantic jealousy due to societal emphasis, there are shared aspects of jealousy in friendships [2]. Exploring jealousy in friendships is essential, as emotional experiences and mechanisms are not exclusive to romantic relationships [8]. This broader understanding contributes to comprehensive theories of jealousy, applicable beyond romance. Recognizing this relevance informs practical interventions in counseling, education, and relationship management. In essence, acknowledging jealousy’s broader significance enhances our understanding of human emotions and relationships.

Assessing the quality of relationships with friends involves evaluating various factors such as communication, trust, support, and overall satisfaction. Some tools consist of structured questions or assessments to measure these dimensions and provide insight into the strengths and weaknesses of the friendship [9]. Friendships, like any relationship, are dynamic and can evolve over time. Assessing the quality of a friendship allows individuals to reflect on the level of connection, mutual understanding, and satisfaction. It can guide efforts to nurture and enhance the relationship [10]. High-quality friendships contribute to emotional well-being, social support, and a sense of belonging [1]. Research consistently shows that positive social connections have numerous benefits for mental health and overall life satisfaction [11]. Addressing jealousy within the context of assessing relationship quality involves recognizing how jealousy may impact overall satisfaction and fulfillment [12]. This integration is essential for understanding the complex interplay between emotions and the broader quality of the friendship.

According to Krappmann [13], although there are shared cultural aspects in the understanding of friendship across various societies, distinctions exist in the interpretation and purpose of friendship based on cultural differences. Within societies and across genders, the definition of friendship varies due to societal influences. It appears that in (modern) Western societies, close friendships are often regarded as personal connections relatively unaffected by societal norms. Regarding Beer [14], in subsistence economies, where resource distribution is not guaranteed, friendships tend to be more instrumentally oriented, focusing on material exchanges.

To assess measurement invariance, researchers evaluate the fit between the specified model and the observed data [15]. For valid comparisons of relationship quality and jealousy between different groups, the measures employed should tap into the same underlying constructs. When this alignment occurs, it indicates measurement invariance [16]. However, some authors disagree that measurement invariance (MI) is necessary for valid comparisons between groups [17,18]. Funder and Gardiner [17] contend that recent discoveries challenging the widespread assumption of profound cultural differences suggest that presuming substantial cross-cultural differences in measurement should not be the default stance. Furthermore, the authors recommend a transition toward external validity as a more meaningful measure of measurement quality. As per Welzel and Inglehart [18], constructs may not necessarily converge at the individual level but can still demonstrate significant and impactful associations at the aggregate level. Hence, the authors propose a shift in paradigm, emphasizing external linkage over internal convergence as the primary criterion for validity.

Assessing the invariance of jealousy and relationship instruments across different groups is important for several reasons, namely, cross-group comparisons (to ensure the validity of any comparisons made between groups, it is crucial that the instruments used to measure jealousy and relationship quality are invariant. If the measures are not invariant, observed differences across groups may be a result of measurement discrepancies rather than true variations in jealousy or relationship quality) [19]; cultural and contextual sensitivity (different cultural or contextual groups may interpret and express emotions like jealousy differently. By assessing measurement invariance, researchers can determine whether the instruments capture the same psychological constructs across diverse groups. This ensures that the instruments are sensitive to cultural or contextual variations in the experience and expression of jealousy and relationship quality) [20]; generalizability of findings (ensuring measurement invariance enhances the generalizability of research findings. If instruments are not invariant, it becomes challenging to extend research conclusions beyond the specific group in which the study was conducted. Invariant instruments allow for more confident generalizations to broader populations) [21]; avoiding biased comparisons (invariance testing helps prevent biased comparisons between groups. If the instruments measure different aspects for different groups, any observed group differences may be a result of measurement bias rather than a difference in jealousy or relationship quality); the validity of constructs (ensuring invariance provides evidence for the validity of the underlying constructs being measured. If the instruments are invariant, it suggests that the same psychological constructs are being assessed consistently across different groups, reinforcing the robustness of the measures) [22]. In research or clinical settings, understanding the invariance of jealousy and relationship measures can have practical implications. It helps researchers and practitioners use these instruments confidently across diverse populations, ensuring that the assessments are meaningful and relevant [23].

## 2. Materials and Methods

### 2.1. Procedures

All procedures conducted throughout this research adhered to the principles outlined in the Declaration of Helsinki and its guidelines governing research involving human subjects. Additionally, the research received approval from the Scientific Council of Catholic Portuguese University. Participants provided informed consent, wherein they were briefed on the study’s objectives, the voluntary nature of their involvement, and the assurance of data anonymity and confidentiality.

A dedicated social media page was created featuring a link for participation to disseminate information about the study. The research protocol encompassed a sociodemographic questionnaire, the Portuguese version of the Multidimensional Jealousy Scale (MJS), and the Portuguese version of the Quality of Relationships Inventory (Friend) (QRI-F). Employing a non-probability approach, specifically snowball sampling, participants were recruited.

Inclusion criteria mandated participants to be 18 years or older, possess Portuguese nationality, and have a minimum educational level enabling comprehension and response to the study questionnaire. Exclusion criteria encompassed failure to meet inclusion criteria and incomplete questionnaire submissions.

### 2.2. Sample

The study comprised 662 participants, with a predominant representation of females, accounting for 73.4% (486 individuals). The average age of the participants was 23.58 years, with a standard deviation of 8.13 (Min = 18, Max = 62). A significant portion of the sample, 71.1% (471 individuals), did not pursue university studies, while the remainder held a university degree. Additionally, a majority of the participants, constituting 75.7% (501 individuals), reported being in a romantic relationship.

### 2.3. Instruments

#### 2.3.1. Sociodemographic Questionnaire

The survey includes inquiries about gender (male/female), age, educational background, and a final question addressing whether participants were currently in a romantic relationship when responding to the questionnaire.

#### 2.3.2. Multidimensional Jealousy Scale (MJS)

The Multidimensional Jealousy Scale (MJS), developed by Pfeiffer and Wong in 1989 [24] and adapted into Portuguese by Lucas et al. [25], is a self-report instrument comprising 17 items. It gauges cognitive, emotional, and behavioral aspects of jealousy, organized into three subscales: cognitive jealousy (assesses the frequency of subjects experiencing worries or irrational thoughts related to jealousy. For instance, Item 2 probes concerns like, “I am worried that someone of the opposite sex is stalking my partner.”); emotional jealousy (evaluates the emotional reactions triggered by situations inducing jealousy. For example, Item 17 explores responses to scenarios where “Your partner comments to you about how attractive someone of the opposite sex is.”); and behavioral jealousy (measures the frequency of engaging in behaviors driven by jealousy. Item 9, for instance, inquires about actions such as, “I go through my partner’s drawers, folders, or pockets.”). Participants rate items on a scale from 1 (never) to 5 (always) for cognitive and behavioral jealousy subscales and from 1 (very good) to 5 (very bad) for the emotional jealousy subscale. A higher scale score indicates a greater perceived level of jealousy. The Portuguese version demonstrated strong internal consistency across all sub-dimensions, with Cronbach’s alpha values of 0.92 for cognitive jealousy, 0.86 for emotional jealousy, 0.90 for behavioral jealousy, and 0.86 for the total scale. This instrument has been validated for different populations, including the Urdu population [26], Italian population [27], Iranian population [28], Serbian population [29], and Australian population [30].

#### 2.3.3. Quality of Relationships Inventory-Friend Version (QRI-F)

The Quality of Relationships Inventory-Friend Version (QRI-F), developed by Pierce et al. [31] and adapted into Portuguese by Neves & Pinheiro [32], is a self-report scale comprising 24 items. This instrument assesses an individual’s perception of support, conflict, and the depth experienced in a specific relationship, specifically focusing on friendships. The scale encompasses three subscales: conflict (gauges how the individual perceives the relationship as a source of conflict and ambivalence. For instance, Item 23 probes how often the person makes you feel angry); support (evaluates the perception of social support from a particular friend. For example, Item 1 assesses the extent to which you can seek advice from this person about various problems); depth (measures the perception of the depth and importance of the friendship. Item 11, for instance, asks how important this relationship is in your life. Participants respond to items on a 4-point Likert scale (1 = Not at all; 2 = A little; 3 = Quite a bit; 4 = Very Much). The Portuguese version demonstrated strong internal consistency, with a Cronbach’s alpha value of 0.88 for the conflict subscale and 0.84 for both the support and depth subscales.

Further studies conducted with samples representing diverse ethnic and cultural backgrounds have yielded consistent findings comparable to the initial validation [32,33,34,35]. In contrast, certain investigations employing similarly diverse samples and various relationship types have revealed a two-factor solution, indicating a departure from the originally proposed factor structure of the Quality of Relationship Inventory (QRI) [36,37].

### 2.4. Data Analysis

The data analysis primarily employed a factor analytic approach using AMOS, treating the items as continuous variables. The weighted least squared means and variance adjusted (WLSMV) method (designed for ordinal data, according to Li [38]) was applied for estimation. The goodness of fit was evaluated using various indices, including chi-square (χ), Root Mean Square Error of Approximation (RMSEA), Standardized Root Mean Square Residual (SRMR), Comparative Fit Index (CFI), Incremental Fit Index (IFI), Goodness of Fit Index (GFI), Tucker–Lewis Index (TLI), and PCLOSE.

For the Multidimensional Jealousy Scale (MJS), multiple analyses were conducted, starting with a unidimensional confirmatory factor analysis. Subsequently, a three-factor confirmatory factor analysis was performed, followed by a second-order three-factor confirmatory factor analysis. Additional analyses included a unidimensional confirmatory factor analysis without item 18, a three-factor confirmatory factor analysis, and a second-order three-factor confirmatory factor analysis without item 18.

Regarding the Quality of Relationships Inventory-Friend Version (QRI-F), three models were tested: initially, a unidimensional confirmatory factor analysis, followed by a three-factor confirmatory factor analysis, and ultimately, a second-order three-factor confirmatory factor analysis.

Measurement invariance was scrutinized using structural equation modeling (SEM) within the framework [16]. Various fit statistics, as recommended by Kline [39], were employed to evaluate the model fit. Configural, metric, scalar, and error invariance, assessing the overall model fit, were evaluated through the chi-square (χ^2^), Root Mean Square Error of Approximation (RMSEA), Standardized Root Mean Square Residual (SRMR), Comparative Fit Index (CFI), Tucker–Lewis Index (TLI), and McDonald’s Noncentrality Index (McNCI). The fit of two nested models was compared to assess metric, scalar, and residual invariance models, and fit statistics for the two models (Δχ^2^, ΔCFI) were presented. Consistent with the literature, a change equal to or below 0.01 was considered the most widely accepted criterion for ensuring invariance [22]. However, for sample sizes with adequate power, Chen [22] proposed an additional criterion: a 0.01 change in CFI, accompanied by changes in RMSEA of 0.015 and SRMR of 0.030 (for metric invariance) or 0.015 (for scalar or residual invariance). The variables employed for assessing measurement invariance underwent categorization: Age was divided into two groups—older and younger—with the cutoff point established at the mean value plus the standard deviation. Likewise, education was segregated into two categories: those with and without university studies. As for relationship status, this variable inherently operated in a dichotomous manner, signifying whether participants were presently engaged in a romantic relationship at the time of questionnaire completion.

Achieving complete measurement invariance in all four steps can be challenging. Therefore, unconventional practices, such as avoiding the constraint of one or more loadings, may be considered to attain partial invariance. As suggested by Steenkamp and Baumgartner [40] and Vandenberg and Lance [21], achieving partial invariance may be acceptable if more than half of the items on a factor remain invariant.

Also, correlations were examined, as well as Cronbach’s alpha (with a criterion of ≥0.70), McDonald’s omega (with a criterion of ≥0.70), composite reliability (CR, with a criterion of ≥0.70), average variance extracted (AVE, with a criterion of ≥0.50), square roots of AVE (with a criterion of ≥0.70), mean, and standard deviation of both the Multidimensional Jealousy Scale (MJS) and the Quality of Relationships Inventory-Friend Version (QRI-F).

In order to identify the factors that contribute to explaining jealousy, a multiple linear regression analysis was conducted. The reported indicators encompassed R-squared and adjusted R-squared values, where higher R-squared values closer to 1 suggest a superior model fit. Additionally, attention was given to *p*-values for predictor variables (considered significant at ≤0.05), unstandardized regression coefficients (B), the unstandardized error of B (EP B), standardized regression coefficients (β), and the F-change statistic along with its significance (*p* ≤ 0.05). These measures collectively provided insights into the strength, significance, and overall impact of the variables in the regression model on the phenomenon of jealousy.

Moderation analyses were conducted using the PROCESS macro to explore potential sociodemographic factors that might influence the relationship between the Quality of Relationships Inventory (Friend) and the Multidimensional Jealousy Scale. The analysis included the presentation of various crucial indices and statistics commonly used to evaluate moderation effects. These included the examination of the interaction effect, characterized by the unstandardized coefficient, standard error, and *p*-value associated with the interaction term. Additionally, the analysis encompassed the assessment of simple slopes, revealing the impact of the independent variable on the dependent variable at different levels of the moderator (e.g., high and low). Conditional effects, indicating the influence at specific values or percentiles of the moderator, were also explored. Furthermore, the study presented the R-squared change (ΔR²), representing the proportion of variance in the dependent variable explained by the interaction. These analyses aimed to provide a comprehensive understanding of how sociodemographic variables might moderate the relationship between relationship quality and jealousy, with a focus on key statistical indices.

Finally, a *t*-test was performed to assess the differences in scores for jealousy and the quality of relationships (friends) between genders. The results included the *t*-test value, its significance level, and the effect size, quantified by Cohen’s d. The statistical analysis software programs used throughout the text were SPSS and AMOS, both in version 28.

## 3. Results

### 3.1. Multidimensional Jealousy Scale

#### 3.1.1. Confirmatory Factorial Analysis

Different models of the Multidimensional Jealousy Scale were tested, with the original model proposed by the authors being the one that presented the best fit. Thus, 17 items and three first-order factors (cognitive jealousy, behavior jealousy, and emotional jealousy) correlated between them were the appropriate structure. It is important to note that the version validated for the Portuguese population contained 18 items, but one of the items (precisely item 18) in this study presented a standardized regression weight of 0.11 and was therefore removed. Furthermore, to achieve a good model, it was necessary to establish correlations between the errors of nine items within the same factors and, therefore, theoretically supported (Table 1, Figure 1).

#### 3.1.2. Testing Invariance

Multigroup CFAs of the Multidimensional Jealousy Scale across gender, age, education, and being in a romantic relationship were carried out. Full configural, metric, scalar, and error variance invariance were achieved for gender, age, education, and being in a romantic relationship (Table 2). However, concerning age, the difference between scalar and error invariance was ΔCFI = 0.014, slightly above the recommended value of 0.010. Thus, only partial error invariance was achieved.

#### 3.1.3. Assessing Reliability

Correlations, Cronbach’s alpha, McDonald’s omega, composite reliability, average variance extracted (AVE), AVE square roots, and mean and standard deviation of the Multidimensional Jealousy Scale were assessed. All values were within the reference ones (Table 3).

### 3.2. Quality of Relationships Inventory (Friend)

#### 3.2.1. Confirmatory Factorial Analysis

Various models of the Quality of Relationships Inventory (Friend) were examined, including a unidimensional factor structure, a three-factor structure, and a second-order model with three factors. The model originally suggested by the authors emerged as the best-fitting one. Consequently, the optimal structure comprised 24 items and three first-order factors (support, conflict, and depth) that exhibited correlated relationships. It was essential to introduce correlations between the errors of 10 items within the same factors to enhance the model, aligning with theoretical support (Table 4 and Figure 2).

#### 3.2.2. Testing Invariance

Multigroup Confirmatory Factor Analyses (CFAs) on the Quality of Relationships Inventory (Friend) across different categories such as gender, age, education, and relationship status were conducted. The analyses successfully achieved full configural, metric, scalar, and error variance invariance for each category, as outlined in Table 5.

#### 3.2.3. Assessing Reliability

Correlations, Cronbach’s alpha, McDonald’s omega, composite reliability, average variance extracted (AVE), AVE square roots, mean, and standard deviation of the Quality of Relationships Inventory (Friend) were examined. All the calculated values fell within the reference ranges, as presented in Table 6.

### 3.3. Multidimensional Jealousy Scale and Quality of Relationships Inventory (Friend)

#### 3.3.1. Correlations

The total scores of MJS and QRI are positively and significantly correlated, although the correlation value is low. The conflict subscale (QRI-F) exhibits the highest correlation with the total MJS score. Cognitive jealousy (MJS) shows significant correlations with all dimensions of QRI-F (positive correlation with QRI-F total and conflict, and negative correlation with support and depth), with the highest correlation observed between cognitive jealousy (MJS) and conflict (QRI-F). Behavior jealousy (MJS) significantly and positively correlates with QRI-F total and conflict (QRI-F). Lastly, emotional jealousy (MJS) only correlates significantly with conflict (QRI-F), although the correlation is very weak (Table 7).

#### 3.3.2. Multiple Linear Regression Analysis

To identify the factors contributing to the explanation of jealousy, a multiple linear regression analysis was conducted. All models found for each dimension of the MJS exhibit very low explanatory power. The one with the highest value is for cognitive jealousy (with an adjusted R-squared value of 0.114); for this dimension, age (being younger) and, especially, high conflict values contribute (Table 8).

#### 3.3.3. Moderations

Moderation analyses were conducted using the PROCESS macro to investigate potential sociodemographic factors influencing the connection between the Quality of Relationships Inventory (Friend) and the Multidimensional Jealousy Scale. Education moderates the link between QRI-F depth and MJS cognitive jealousy, as well as between QRI-F conflict and MJS cognitive jealousy; the absence of university studies enhances these associations. Relationship status moderates the connection between QRI-F total and MJS behavioral jealousy, as well as the link between QRI-F conflict and MJS behavioral jealousy; being in a romantic relationship strengthens these associations. Lastly, gender moderates the link between QRI-F total and MJS behavioral jealousy, with being a woman intensifying this relationship (Table 9).

#### 3.3.4. Differences

The findings indicate statistically significant gender differences in the MJS subscales of behavioral jealousy and emotional jealousy, as well as in the total score and all subscales of QRI-F. Across these dimensions, females score significantly higher than males, except in the conflict subscale (QRI-F), where males score higher than females (Table 10).

Additionally, significant differences exist between individuals without university studies and those with university education regarding the emotional jealousy subscale (MJS) [*t*(345,616) = −2.385; *p* = 0.016; d = −0.206; SE = 0.086] and the depth subscale (QRI-F) [*t*(412, 971) = −2.557; *p* = 0.011; d = −0.204; SE = 0.087]: individuals with university studies exhibit higher values (MJS, M = 16.17; SD = 7.41; QRI-F, M = 3.35; SD = 0.50) compared to those without university studies, in both dimensions (MJS, M = 14.67; SD = 7.27; QRI-F, M = 3.23; SD = 0.60).

Also, there are statistically significant differences between individuals who are not in a romantic relationship and those who are, specifically concerning the emotional jealousy subscale (MJS) [*t*(295, 308) = −2.200; *p* = 0.029; d = 0.189; SE = 0.091] and the behavior jealousy subscale (MJS) [*t*(288, 208) = −2.557; *p* = 0.011; d = 0.223; SE = 0.091], with those who are in a romantic relationship presenting higher values (MJS-EJ, M = 15.44; SD = 7.48; MJS-BJ-F, M = 7.60; SD = 2.68) than those who are not (MJS-EJ, M = 14.05; SD = 6.78; MJS-BJ-F, M = 7.09; SD = 2.50) in both the dimensions.

Age correlates significantly and negatively with QRI-F total (r = −0.128; SE = 0.039; *p* < 0.001), QRI-F support (r = −0.204; SE = 0.038; *p* < 0.001), and QRI-F depth (r = −0.146; SE = 0.039; *p* < 0.001).

## 4. Discussion

The aim of this study was to investigate the consistency of the Multidimensional Jealousy Scale (MJS) and the Quality of Relationships Inventory (Friend) (QRI-F) across various demographic factors in a Portuguese sample. Confirmatory factor analysis (CFA) was employed to assess the suitability of different factor structures. The MJS was best represented by the original model proposed by the authors (original version: Pfeiffer & Wong [24]; Portuguese version: Lucas et al. [25]), featuring 17 items and three correlated first-order factors (cognitive jealousy, behavior jealousy, and emotional jealousy). Similarly, the optimal structure for the QRI-F consisted of 24 items and three correlated first-order factors (support, conflict, and depth) as proposed by the original authors (original version: Pierce et al. [31]; Portuguese version: Neves & Pinheiro [32]). CFA is a critical tool in research, serving various purposes such as testing and evaluating theoretical models, establishing construct validity, identifying measurement errors, assessing factor loadings, comparing alternative models, testing hypotheses, cross-validating findings across different samples, and refining measurement models [41]. This statistical technique ensures that measurement instruments accurately capture the intended constructs and helps researchers make informed decisions about the appropriateness and precision of their models [20].

Multigroup Confirmatory Factor Analyses (CFAs) were conducted on the Multidimensional Jealousy Scale and the Quality of Relationships Inventory (Friend) (QRI-F), examining potential variations across gender, age, education, and relationship status. Full configural, metric, scalar, and error variance invariance were successfully achieved for all these demographic variables. This means that the Multidimensional Jealousy Scale (MJS) and Quality of Relationships Inventory (QRI-F) provide assessments of romantic jealousy and relationship quality that are equivalent and consistent across gender, age, education, and relationship status. The importance of performing multi-group analysis in research, especially in the social sciences, lies in several key reasons. These include the need for assessing measurement invariance ensuring that measurement properties are consistent across diverse groups to interpret constructs similarly [21]. It allows for comparing groups to test whether relationships between variables hold consistently across different groups. Multi-group analysis is essential for examining group differences in structural models, aiding in understanding how these relationships may vary. In cross-cultural research, it ensures that measurements are equivalent across different cultural groups [42]. Additionally, it has implications for policy and interventions by tailoring strategies to specific demographic or cultural groups. Failing to conduct a multi-group analysis may introduce biases, as assumptions about the equality of measurement or structural parameters across groups could be violated. Ultimately, performing multi-group analysis enhances construct validity by ensuring consistent measurement of the same construct across diverse groups [43]. However, some authors consider that the selection of the measurement invariance method appears to lack a clear rationale. In theory, local structural equation modeling (LSEM) [44] and moderated nonlinear factor analysis (MNLFA) [45,46] are considered more appropriate methodologies when compared to the use of multiple-group confirmatory factor analysis.

In this study, it was thoroughly examined correlations, as well as the reliability of measurements assessed through Cronbach’s alpha, McDonald’s omega, and composite reliability (commonly applied in Structural Equation Models). Additionally, convergent validity was evaluated using the average variance extracted (AVE) and discriminant validity through AVE square roots for both the Multidimensional Jealousy Scale and the Quality of Relationships Inventory (Friend) [47]. All these values fell within the established reference ranges. Consequently, the reliability, reflecting the consistency, stability, or repeatability of the measures employed in this study, was confirmed. Moreover, we verified convergent validity, indicating the degree to which different measures that should theoretically be related do indeed exhibit a relationship [39]. Furthermore, discriminant validity, assessing the extent to which a measure does not strongly correlate with measures of different, unrelated constructs, was established. These results collectively affirm the robust psychometric qualities of the instruments used in our study [48].

The overall scores of the Multidimensional Jealousy Scale (MJS) and the Quality of Relationships Inventory (Friend) (QRI) show a positive and statistically significant correlation, albeit with a relatively low magnitude. Notably, the conflict subscale of the Quality of Relationships Inventory (QRI-F) demonstrates the strongest correlation with the total score on the Multidimensional Jealousy Scale (MJS). Specifically, the most pronounced correlation is observed between cognitive jealousy (MJS) and conflict (QRI-F). The correlation between jealousy and the quality of friendships can vary, influenced by individual differences, the dynamics of the relationship, and the specific context under examination. In healthy friendships, a typical pattern involves a negative correlation between jealousy and friendship quality [49]. Elevated levels of jealousy often signal lower friendship quality, suggesting potential feelings of mistrust, competition, or insecurity within the friendship [2]. Friendships characterized by trust, open communication, and mutual support tend to exhibit lower levels of jealousy. When individuals feel secure in their friendship, expressions of jealousy are less likely [50].

However, in certain scenarios, especially those involving competition or perceived threats to friendship, jealousy may be more prevalent. For instance, if friends are competing for the same opportunities or vying for the attention of a mutual friend, jealousy might be heightened [51]. Individual differences, such as personality traits and attachment styles, play a role in shaping the correlation between jealousy and friendship quality. Individuals with higher levels of insecurity or possessiveness may be more susceptible to jealousy, potentially impacting the overall quality of their friendships [52]. How individuals and friends navigate conflicts related to jealousy can also influence overall friendship quality. Friends adept at resolving conflicts, addressing concerns, and maintaining open communication are more likely to sustain high-quality friendships, even in situations where jealousy may occasionally arise [53].

All identified models, focusing on each dimension of the Multidimensional Jealousy Scale (MJS), exhibit notably low explanatory power. Among these, the model with the highest explanatory value pertains to cognitive jealousy. In this dimension, factors such as age (being younger) and notably elevated conflict values play significant roles. The presence of conflict within a relationship is closely associated with or contributes to the manifestation of cognitive jealousy. Cognitive jealousy typically involves persistent thoughts, worries, or irrational suspicions regarding a partner’s actions, intentions, or interactions with others [24]. Conflict is identified as a contributing factor to cognitive jealousy through several mechanisms, namely, trust issues (compromised trust can render individuals more susceptible to cognitive jealousy, evoking feelings of insecurity, fear of rejection, abandonment, or inadequacy) [54]; communication breakdown (lack of clear communication or misunderstandings during conflicts may lead individuals to misinterpret their partner’s actions, thereby contributing to cognitive jealousy) [55]; resurfacing of past issues (previous experiences of betrayal or hurt may resurface during conflicts, intensifying cognitive jealousy as individuals project past experiences onto current situations) [1]; emotional turmoil (emotional distress during conflicts may result in heightened vigilance, suspicion, or irrational thoughts associated with cognitive jealousy) [56]; and perceived threats (conflict may trigger cognitive jealousy by making individuals hyper-aware of potential threats to their emotional connection, further fueling suspicions and concerns) [57].

The findings reveal significant gender-based differences in the Multidimensional Jealousy Scale (MJS) subscales, specifically in behavioral jealousy and emotional jealousy, as well as in the total scores and all subscales of the Quality of Relationships Inventory-Friend Version (QRI-F). Across these dimensions, females consistently score significantly higher than males, except for the conflict subscale (QRI-F), where males score higher than females. The observed pattern aligns with established trends where women frequently report elevated levels of emotional jealousy, reflecting concerns about their partner forming emotional connections with others. This heightened sensitivity may lead women to experience greater jealousy when confronted with emotional closeness between their partner and others [58]. Conversely, men, on average, tend to report higher levels of sexual jealousy, emphasizing concerns about their partner’s physical infidelity. Men may be more susceptible to jealousy in response to perceived sexual threats or infidelity [59]. Evolutionary psychology offers a perspective suggesting that these gender differences may be linked to distinct reproductive strategies. Men’s heightened sensitivity to sexual infidelity may stem from concerns about paternity, while women may prioritize concerns about emotional infidelity to ensure investment in offspring [60,61]. Social and cultural factors further contribute to shaping gender differences in jealousy. Norms and expectations related to gender roles and relationships play a role in influencing how jealousy is expressed and perceived within different cultural and societal contexts [58].

In line with our findings, prior research by Demir and Orthel [62] supports the notion that women tend to experience higher-quality and less conflicted real and ideal best friendships compared to men. This aligns with our observed patterns. Additionally, Greif [63] suggests that men and women have distinct friendship needs, with men seeking a diverse network of friends and women needing must friends (closest and best friends) and trust friends (reliable friends). Our results echo this understanding. Moreover, our study contributes to the existing literature by revealing that gender plays a moderating role in the connection between friendship quality and behavioral jealousy. Specifically, our findings indicate that being a woman enhances the strength of this relationship, implying that women may be more susceptible to behavioral jealousy in response to variations in the quality of their friendships. This aligns with broader research suggesting that gender differences influence the dynamics of friendships and emotional responses, particularly in the context of jealousy [64].

Individuals with university studies exhibit statistically significant differences compared to those without such qualifications, particularly in the realms of emotional jealousy (MJS) and depth (QRI-F). Those with university studies tend to score higher in both dimensions. The higher values observed in individuals with university studies in depth could be attributed to several factors linked to higher education. For instance, increased formal education may correlate with enhanced communication skills [65]. Proficient communication is pivotal in managing and expressing emotions, including navigating feelings of jealousy in a relationship [66].

Additionally, formal education may contribute to the development of emotional intelligence, aiding individuals in understanding and regulating their emotions, potentially influencing their experience and handling of jealousy [67]. Moreover, higher education is often associated with improved problem-solving and critical-thinking skills. This cognitive prowess may positively impact how individuals approach and resolve issues related to jealousy within their relationships [68].

Furthermore, education plays a moderating role in the relationship between QRI-F depth and MJS cognitive jealousy, as well as between QRI-F conflict and MJS cognitive jealousy. Not having university studies strengthens these connections. The observation that individuals with higher levels of education tend to experience more jealousy might seem counterintuitive, as one might expect greater education to be associated with emotional maturity and enhanced interpersonal skills. However, several factors could contribute to this seemingly paradoxical finding: increased expectations (individuals with higher education may have higher expectations for their relationships and may be more aware of potential threats or challenges [69]. This heightened awareness could contribute to increased feelings of jealousy); complexity of relationships (higher education may expose individuals to a variety of social situations and relationship dynamics. This exposure could lead to a greater awareness of the complexities and nuances within relationships, potentially contributing to heightened emotional responses, including jealousy) [70]; striving for success (individuals with more education may be ambitious and goal-oriented, striving for success in various aspects of life, including relationships. The fear of losing a partner or perceived threats to the relationship’s success could lead to increased feelings of jealousy) [71]; communication challenges (while higher education can enhance communication skills, it might also lead to overthinking or misinterpreting communication cues. The ability to critically analyze situations may sometimes result in reading too much into behaviors, triggering jealousy) [65]; comparison with peers (individuals with higher education may engage in more social comparisons, comparing their relationships and achievements with those of their peers. This comparative mindset could contribute to feelings of inadequacy or competition, fostering jealousy) [71]; and perfectionism (higher education may be associated with higher standards or perfectionist tendencies. Individuals with perfectionist traits might be more prone to jealousy when they perceive a discrepancy between their idealized expectations and the reality of their relationships) [72].

Differences were observed between individuals in romantic relationships and those not in romantic relationships concerning both the emotional and behavioral jealousy subscales of the Multidimensional Jealousy Scale (MJS). Individuals in romantic relationships tended to exhibit higher scores on both dimensions compared to those not in romantic relationships. The relationship status played a moderating role in the connection between the total score of the Quality of Relationships Inventory (Friend) (QRI-F) and behavioral jealousy in the MJS, as well as the association between conflict in QRI-F and behavioral jealousy in MJS; being in a romantic relationship strengthened these connections. This finding aligns with insights from White and Mullen [73], who proposed that commitment in a romantic relationship could potentially reduce jealousy by providing a sense of security for investments and self, along with minimizing the likelihood of a rival relationship [73], p. 111. Additionally, research by Worley and Samp [2] suggests that a partner’s involvement in cross-sex friendships could evoke perceptions of threat to both the existence and quality of the romantic relationship. The specific forms of jealousy experienced were found to be influenced by individuals’ threat appraisals associated with these friendships, including dimensions such as sexual jealousy, companionship jealousy, intimacy jealousy, power jealousy, and relational quality threat [2].

Age exhibits significant and negative correlations with the total score of the Quality of Relationships Inventory-Friend Version (QRI-F), as well as its subscales: QRI-F support and QRI-F depth. However, these findings diverge from existing literature on the topic, which suggests that gender differences play a role in how friendship quality evolves during emerging adulthood. Contrary to the common patterns reported in the literature, this study did not align with the notion that women typically experience an increase in overall friendship quality, emotional support, and intimacy but a decrease in instrumental support during emerging adulthood. For men, emotional support is expected to remain stable, while intimacy and instrumental support tend to increase [74]. Moreover, some research indicates a general decline in friendship quality during this life stage, with the exception of specific features such as companionship and reliable alliance, which may even strengthen in the early twenties [74]. The observed discrepancies in this study could be attributed to various factors, including life stage transitions such as career commencement, family-raising, or entering retirement. Additionally, communication patterns tend to evolve with age, influencing relationship satisfaction. Effective communication skills are often associated with positive relationship outcomes. Generational or cohort effects, reflecting the values and societal norms of a specific generation, might also contribute to the variations in relationship dynamics observed across age groups. Changes in societal attitudes toward relationships and marriage can vary across generations, further shaping the nature of interpersonal connections.

### 4.1. Limitations

This study is subject to several acknowledged limitations. Firstly, the non-representative nature of the sample in relation to the broader Portuguese population impedes the generalization of the findings. Furthermore, the convenience sampling method employed resulted in a predominantly female composition, potentially introducing bias to the results. Another notable limitation is the scarcity of research on jealousy within friendship relationships, rendering the literature search challenging. Additionally, the reliance on self-report instruments introduces subjectivity to the measurement process. The utilization of a cross-sectional design offers a singular snapshot in time, constraining the ability to establish causal relationships. The issue of cultural specificity is also relevant, as the results may not be universally applicable across diverse cultural or contextual settings.

### 4.2. Future Research

In future research focusing on jealousy in friendship relationships, there is a potential for advancing the field through several key avenues. Firstly, researchers may consider refining and validating instruments, particularly the Multidimensional Jealousy Scale. The aim would be to improve its reliability and validity, allowing for a more nuanced and accurate measurement of jealousy within the context of friendships. Additionally, exploring further dimensions or factors that contribute to jealousy in friendships could be a valuable endeavor, ensuring a comprehensive understanding of this intricate emotion. Moreover, investigating how various aspects of friendship, including trust, communication, and reciprocity, correlate with and influence experiences of jealousy presents another promising area for research. Understanding the dynamics between these friendship qualities and jealousy could provide valuable insights into the complex interplay of emotions within friendships. Cultural and contextual variations in the experience and expression of jealousy within friendships could also be a significant focus. By examining how societal norms and expectations impact the perception and management of jealousy among friends, researchers can uncover insights into the diverse ways this emotion manifests across different cultural and social contexts.

Overall, delving into these research directions has the potential to deepen our understanding of jealousy in friendships, offering insights that can inform interventions and contribute to the broader field of interpersonal relationships.

## 5. Conclusions

This study aimed to investigate the Multidimensional Jealousy Scale (MJS) and the Quality of Relationships Inventory (Friend) (QRI-F) across demographic factors in a Portuguese sample. Confirmatory factor analysis (CFA) assessed factor structures, confirming the original MJS model and the QRI-F structure. Multigroup CFAs explored variations across gender, age, education, and relationship status, achieving configural, metric, scalar, and error variance invariance. The study highlighted the importance of multi-group analysis in social sciences for assessing measurement invariance and understanding relationships across diverse groups. The research ensured the reliability and convergent and discriminant validity of the instruments. Overall scores of MJS and QRI-F showed a positive correlation, with conflict (QRI-F) exhibiting the strongest correlation with cognitive jealousy (MJS). The correlation between jealousy and friendship quality varied, influenced by individual differences, relationship dynamics, and context. The study explored correlations with demographic factors, revealing gender-based differences and education’s moderating role in jealousy. Women consistently scored higher in emotional jealousy, aligning with established trends. Education’s impact on jealousy was nuanced, suggesting increased expectations, relationship complexities, and communication challenges. Differences between individuals in romantic relationships and those not revealed higher jealousy scores in the former, emphasizing commitment’s potential role in reducing jealousy. Age exhibited negative correlations with friendship quality, contrary to the literature, potentially influenced by life stage transitions, evolving communication patterns, generational effects, and changing societal attitudes toward relationships. The study provided comprehensive insights into jealousy, emphasizing its multifaceted nature across demographic factors and shedding light on the intricacies of friendship dynamics.

## Figures and Tables

**Figure 1 behavsci-14-00044-f001:**
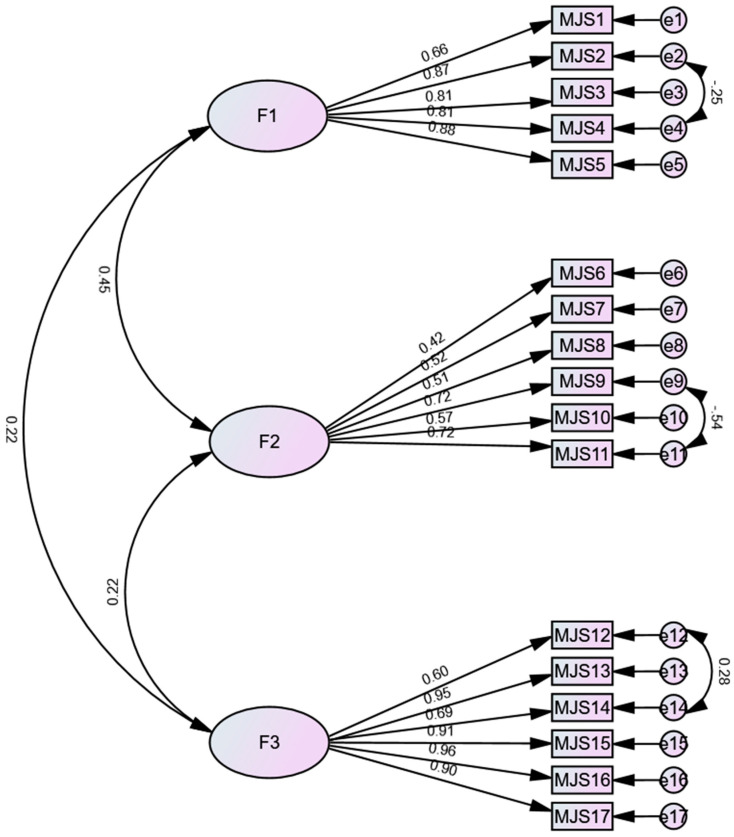
The final CFA model of the Multidimensional Jealousy Scale.

**Figure 2 behavsci-14-00044-f002:**
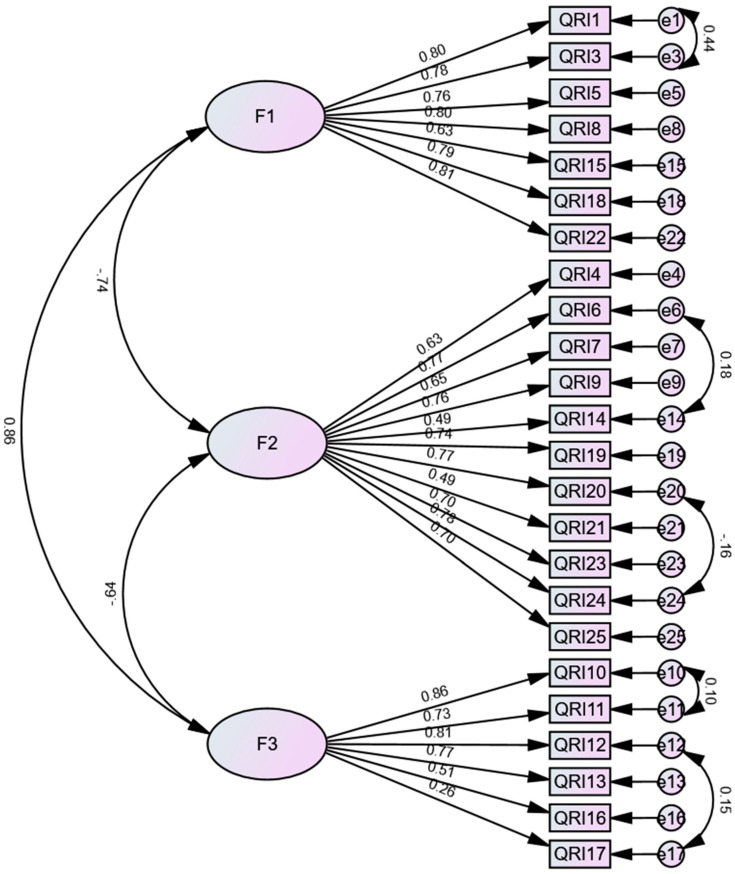
The final CFA model of the Quality of Relationships Inventory.

**Table 1 behavsci-14-00044-t001:** Multidimensional Jealousy Scale fit indices for total sample.

	Fit Indices of Models										
	χ^2^	df	χ^2^/df	*p*	CFI	IFI	GFI	TLI	RMSEA (90%CI)	PCLOSE	SRMR
1 factor 18 items	854.851	136	6.286	0.000	0.638	0.641	0.821	0.592	0.089 (0.084–0.095)	0.000	0.247
3 factors	547.761	132	4.150	0.000	0.790	0.793	0.885	0.757	0.069 (0.063–0.075)	0.000	0.088
Second order 3 factors	570.537	134	4.258	0.000	0.780	0.782	0.881	0.749	0.070 (0.064–0.076)	0.000	0.100
1 factor without item 18	734.494	120	6.121	0.000	0.649	0.652	0.813	0.602	0.088 (0.082–0.094)	0.000	0.237
3 factors without item 18	429.434	116	3.702	0.000	0.821	0.823	0.891	0.790	0.064 (0.058–0.070)	0.000	0.076
Second order 3 factors without item 18	451.706	118	3.828	0.000	0.809	0.811	0.885	0.780	0.065 (0.059–0.072)	0.000	0.089
	Fit indices of the chosen model ^1^										
3 factors without item 18	245.411	107	2.294	0.000	0.921	0.922	0.938	0.900	0.044 (0.037–0.052)	0.901	0.062

*Note*: ^1^ Fit indices were adjusted after residuals correlations of 6 items; *p* < 0.001 for all indicators; χ^2^ = chi-square; df = degrees of freedom; CFI = comparative fit index; IFI–incremental fit index; GFI = goodness of fit index; TLI = Tuck-Lewis index; RMSEA = root mean square error of approximation; CI = confidence interval; SRMR = standardized root mean square residual.

**Table 2 behavsci-14-00044-t002:** Multigroup CFAs of the Multidimensional Jealousy Scale across gender, age, education, and being in a romantic relationship.

Gender	χ^2^	df	χ^2^/df	RMSEA (CI)	CFI	IFI	SRMR	Comparisons	ΔRMSEA	ΔCFI	ΔSRMR	Δχ^2^/df
Configural invariance	607.416	214	2.838	0.053 (0.048–0.058)	0.961	0.961	0.065	NA	NA	NA	NA	NA
Metric invariance	831.618	228	3.647	0.063 (0.059–0.068)	0.94	0.94	0.071	Configural vs. metric	0.010	0.003	0.006	0.809
Scalar invariance	1023.878	234	4.376	0.072 (0.067–0.076)	0.921	0.922	0.083	Metric vs. scalar	0.009	0.003	0.012	.0729
Error variance invariance	1330.541	260	5.117	0.079 (0.075–0.083)	0.891	0.894	0.088	Scalar vs. error variance	0.007	0.010	0.005	0.741
**Age**	**χ^2^**	**df**	**χ^2^/df**	**RMSEA (CI)**	**CFI**	**IFI**	**SRMR**	**Comparisons**	**ΔRMSEA**	**ΔCFI**	**ΔSRMR**	**Δχ^2^/df**
Configural invariance	504.225	214	2.356	0.045 (0.040–0.050)	0.904	0.906	0.059	NA	NA	NA	NA	NA
Metric invariance	620.613	228	2.722	0.051 (0.046–0.056)	0.903	0.904	0.060	Configural vs. metric	0.006	0.001	0.001	0.366
Scalar invariance	630.588	234	2.695	0.051 (0.046–0.055)	0.902	0.903	0.063	Metric vs. scalar	0.000	0.001	0.003	0.027
Error variance invariance	771.654	260	2.968	0.055 (0.050–0.059)	0.897	0.899	0.065	Scalar vs. error variance	0.004	0.005	0.002	0.273
**Education**	**χ^2^**	**df**	**χ^2^/df**	**RMSEA (CI)**	**CFI**	**IFI**	**SRMR**	**Comparisons**	**ΔRMSEA**	**ΔCFI**	**ΔSRMR**	**Δχ^2^/df**
Configural invariance	490.992	214	2.294	0.044 (0.039–0.049)	0.940	0.940	0.070	NA	NA	NA	NA	NA
Metric invariance	511.055	228	2.241	0.043 (0.038–0.048)	0.941	0.941	0.069	Configural vs. metric	0.001	0.001	0.001	0.053
Scalar invariance	524.388	234	2.241	0.043 (0.038–0.048)	0.941	0.941	0.069	Metric vs. scalar	0.000	0.000	0.000	0.000
Error variance invariance	611.049	260	2.350	0.045 (0.039–0.048)	0.937	0.937	0.074	Scalar vs. error variance	0.002	0.004	0.005	0.109
**Romantic Relationship**	**χ^2^**	**df**	**χ^2^/df**	**RMSEA (CI)**	**CFI**	**IFI**	**SRMR**	**Comparisons**	**ΔRMSEA**	**ΔCFI**	**ΔSRMR**	**Δχ^2^/df**
Configural invariance	505.494	214	2.362	0.045 (0.040–0.051)	0.934	0.934	0.066	NA	NA	NA	NA	NA
Metric invariance	525.983	228	2.307	0.044 (0.040–0.049)	0.933	0.933	0.068	Configural vs. metric	0.001	0.001	0.002	0.055
Scalar invariance	565.448	234	2.416	0.046 (0.041–0.051)	0.932	0.932	0.068	Metric vs. scalar	0.002	0.001	0.000	0.109
Error variance invariance	823.772	260	3.168	0.056 (0.053–0.059)	0.930	0.930	0.069	Scalar vs. error variance	0.010	0.002	0.001	0.752

*Note*: χ^2^ = qui-squared; df = degrees of freedom; IFI = incremental fit index; CFI = comparative fit index; RMSEA = root mean square error of approximation; CI = confidence interval; SRMS = standard root mean square; ΔRMSEA = change in RMSEA compared with the previous model (expressed in absolute values); ΔCFI = change in CFI compared with the previous model (expressed in absolute values); ΔSRMR = change in SRMR compared with the previous model (expressed in absolute values). All models are significant at *p* < 0.001; NA = not applicable.

**Table 3 behavsci-14-00044-t003:** Correlations, Cronbach’s alpha, McDonald’s omega, composite reliability, average variance extracted (AVE), AVE square roots, and mean and standard deviation of the Multidimensional Jealousy Scale.

	Pearson Correlations								
	0	1	2	3	α	ω	CR	AVE	Mean (*SD*)
0. MJS Total	**0.707**				0.864	0.851	0.883	0.500	33.05 (11.05)
1. Cognitive jealousy	0.691 **	**0.815**			0.873	0.869	0.902	0.665	8.11 (3.93)
2. Behavior jealousy	0.569 **	0.376 **	**0.711**		0.740	0.737	0.829	0.505	7.46 (2.65)
3. Emotional jealousy	0.828 **	0.257 **	0.204 **	**0.868**	0.913	0.924	0.932	0.699	15.10 (7.34)

*Note*: ** *p* < 0.001; α = Cronbach’s alpha; ω = McDonald’s omega; CR = composite reliability; AVE = average variance extracted; **bold** (diagonal) = AVE square roots; *SD* = Standard deviation.

**Table 4 behavsci-14-00044-t004:** Quality of Relationships Inventory (Friend) fit indices for total sample.

	Fit Indices of Models										
	χ^2^	df	χ^2^/df	*p*	CFI	IFI	GFI	TLI	RMSEA (90%CI)	PCLOSE	SRMR
1 factor	1318.405	252	5.232	0.000	0.685	0.684	0.643	0.60	0.080 (0.076–0.084)	0.000	0.091
3 factors	1091.693	249	4.384	0.000	0.879	0.879	0.865	0.873	0.072 (0.067–0.076)	0.000	0.070
Second order 3 factors	1279.801	251	5.099	0.000	0.729	0.730	0.715	0.712	0.079 (0.074–0.083)	0.000	0.088
	Fit indices of the chosen model ^1^										
3 factors	552.337	221	2.499	0.000	0.910	0.916	0.920	0.900	0.048 (0.043–0.053)	0.779	0.040

*Note*: ^1^ Fit indices were adjusted after residuals correlations of 10 items; *p* < 0.001 for all indicators; χ^2^ = chi-square; df = degrees of freedom; CFI = comparative fit index; GFI = goodness of fit index;; TLI = Tuck-Lewis index; RMSEA = root mean square error of approximation; CI = confidence interval; SRMR = standardized root mean square residual.

**Table 5 behavsci-14-00044-t005:** Multigroup CFAs of the Quality of Relationships Inventory (Friend) across gender, age, education, and being in a romantic relationship.

Gender	χ^2^	df	χ^2^/df	RMSEA (CI)	CFI	IFI	SRMR	Comparisons	ΔRMSEA	ΔCFI	ΔSRMR	Δχ^2^/df
Configural invariance	779.101	444	1.755	0.034 (0.030–0.038)	0.937	0.938	0.061	NA	NA	NA	NA	NA
Metric invariance	814.839	465	1.752	0.034 (0.030–0.038)	0.935	0.936	0.067	Configural vs. metric	0.000	0.002	0.005	0.003
Scalar invariance	826.335	471	1.754	0.034 (0.030–0.038)	0.933	0.934	0.070	Metric vs. scalar	0.000	0.002	0.003	0.002
Error variance invariance	910.527	522	1.744	0.034 (0.030–0.037)	0.923	0.924	0.074	Scalar vs. error variance	0.000	0.010	0.004	0.010
**Age**	**χ^2^**	**df**	**χ^2^/df**	**RMSEA (CI)**	**CFI**	**IFI**	**SRMR**	**Comparisons**	**ΔRMSEA**	**ΔCFI**	**ΔSRMR**	**Δχ^2^/df**
Configural invariance	818.252	444	1.843	0.037 (0.033–0.040)	0.927	0.928	0.065	NA	NA	NA	NA	NA
Metric invariance	837.963	465	1.802	0.037 (0.033–0.041)	0.929	0.929	0.067	Configural vs. metric	0.000	0.002	0.002	0.041
Scalar invariance	854.55	471	1.814	0.037 (0.033–0.040)	0.919	0.919	0.073	Metric vs. scalar	0.000	0.010	0.006	0.012
Error variance invariance	1016.387	522	1.947	0.040 (0.036–0.043)	0.909	0.909	0.068	Scalar vs. error variance	0.003	0.010	0.005	0.133
**Education**	**χ^2^**	**df**	**χ^2^/df**	**RMSEA (CI)**	**CFI**	**IFI**	**SRMR**	**Comparisons**	**ΔRMSEA**	**ΔCFI**	**ΔSRMR**	**Δχ^2^/df**
Configural invariance	830.245	444	1.870	0.036 (0.032–0.040)	0.948	0.950	0.064	NA	NA	NA	NA	NA
Metric invariance	853.094	465	1.835	0.036 (0.032–0.049)	0.946	0.949	0.064	Configural vs. metric	0.000	0.002	0.000	0.035
Scalar invariance	884.749	471	1.878	0.036 (0.033–0.040)	0.944	0.947	0.067	Metric vs. scalar	0.000	0.002	0.003	0.043
Error variance invariance	1114.299	522	2.135	0.041 (0.038–0.045)	0.940	0.945	0.069	Scalar vs. error variance	0.005	0.004	0.002	0.257
**Romantic Relationship**	**χ^2^**	**df**	**χ^2^/df**	**RMSEA (CI)**	**CFI**	**IFI**	**SRMR**	**Comparisons**	**ΔRMSEA**	**ΔCFI**	**ΔSRMR**	**Δχ^2^/df**
Configural invariance	869.922	444	1.959	0.038 (0.034–0.042)	0.934	0.936	0.066	NA	NA	NA	NA	NA
Metric invariance	893.62	465	1.922	0.037 (0.034–0.041)	0.932	0.933	0.065	Configural vs. metric	0.001	0.002	0.001	0.037
Scalar invariance	916.209	471	1.945	0.038 (0.034–0.041)	0.930	0.932	0.068	Metric vs. scalar	0.001	0.002	0.003	0.023
Error variance invariance	1033.929	522	1.981	0.039 (0.035–0.042)	0.928	0.931	0.069	Scalar vs. error variance	0.001	0.002	0.001	0.051

*Note*: χ^2^ = qui-squared; df = degrees of freedom; IFI = incremental fit index; CFI = comparative fit index; RMSEA = root mean square error of approximation; CI = confidence interval; SRMS = standard root mean square; ΔRMSEA = change in RMSEA compared with the previous model (expressed in absolute values); ΔCFI = change in CFI compared with the previous model (expressed in absolute values); ΔSRMR = change in SRMR compared with the previous model (expressed in absolute values). All models are significant at *p* < 0.001; NA = not applicable.

**Table 6 behavsci-14-00044-t006:** Correlations, Cronbach’s alpha, McDonald’s omega, composite reliability, average variance extracted (AVE), AVE square roots, and mean and standard deviation of the Quality of Relationships Inventory.

	Pearson Correlations								
	0	1	2	3	α	ω	CR	AVE	Mean (*SD*)
0. QRI Total	**0.707**				0.814	0.816	0.750	0.500	2.72 (0.34)
1. Support	0.583 **	**0.790**			0.892	0.894	0.920	0.624	3.36 (0.58)
2. Conflict	0.555 **	−0.299 **	**0.708**		0.886	0.885	0.908	0.501	2.02 (0.56)
3. Depth	0.693 **	0.749 **	−0.130 **	**0.773**	0.853	0.854	0.897	0.598	3.27 (0.57)

*Note*: ** *p* < 0.001; α = Cronbach’s alpha; ω = McDonald’s omega; CR = composite reliability; AVE = average variance extracted; **bold** (diagonal) = AVE square roots; *SD* = Standard deviation.

**Table 7 behavsci-14-00044-t007:** Pearson correlations between the Multidimensional Jealousy Scale and the Quality of Relationships Inventory (Friend).

	Pearson Correlations			
	MJS Total	Cognitive Jealousy	Behavior Jealousy	Emotional Jealousy
QRI Total	0.163 **	0.155 **	0.176 **	0.075
Support	−0.072	−0.114 **	−0.041	−0.006
Conflict	0.284 **	0.330 **	0.266 **	0.094 *
Depth	−0.039	−0.092 *	−0.012	0.016

*Note*: ** *p* < 0.001; * *p* < 0.01.

**Table 8 behavsci-14-00044-t008:** Sociodemographic and Quality of Relationships Inventory (Friend) variables that contribute to explaining Multidimensional Jealousy Scale.

	MJS Total			Cognitive Jealousy			Behavior Jealousy			Emotional Jealousy		
	B	SE	β	B	SE	β	B	SE	β	B	SE	β
Gender	1.834	0.965	0.073	−0.238	0.337	−0.027	0.603	0.231	0.101	1.265	0.661	0.076
Age	0.015	0.054	0.011	−0.042	0.019	−0.088	0.020	0.013	0.062	0.041	0.037	0.046
Education	0.323	0.276	0.045	0.006	0.096	0.002	−0.056	0.066	−0.033	0.392	0.189	0.082
Romantic relationship	−0.596	0.998	−0.023	0.409	0.349	0.045	−0.352	0.239	−0.057	−0.899	0.684	−0.053
QRI Total	^†^			^†^			^†^			^†^		
Support	0.506	1.160	0.026	0.287	0.405	0.042	0.209	0.278	0.045	−0.009	0.795	−0.001
Conflict	5.735	0.780	0.292	2.339	0.273	0.334	1.325	0.187	0.281	1.280	0.535	0.098
Depth	−0.645	1.110	−0.034	−0.621	0.388	−0.091	−0.055	0.266	−0.012	0.252	0.760	0.020
R^2^ (R^2^ Adj.)	0.086 (0.083)			0.117 (0.114)			0.084 (0.081)			0.025 (0.021)		
F for change in R^2^	60.367 **			82.926 **			54.338 **			6.862 *		

R^2^ = R squared; R^2^ Adj. = R squared adjusted; B = unstandardized regression coefficients; SE = unstandardized error of B; β = standardized regression coefficients; * *p* < 0.010; ** *p* < 0.001; ^†^ excluded from de model.

**Table 9 behavsci-14-00044-t009:** Sociodemographic moderators in the relationship between the Quality of Relationships Inventory (Friend) and the Multidimensional Jealousy Scale.

Predictor	Moderator	Dependent	F(5, 656)	*p*	β	95% CI	*t*	*p*	Variance %	Moderator Option	β	*p*
QRI-F Depth	Education	MJS Cognitive jealousy	2.230	0.049	3.321	0.486, 6.156	2.300	0.022	12.93	No university studies	−0.805	0.008
QRI-F Conflict	Education	MJS Cognitive jealousy	17.295	<0.001	3.016	0.418, 5.613	2.280	0.023	34.13	No university studies	2.108	<0.001
QRI-F Total	Romantic relationship	MJS Behavioral jealousy	12.460	<0.001	−2.157	−3.490, −0.823	−3.175	0.002	23.19	Yes	1.898	<0.001
QRI-F Conflict	Romantic relationship	MJS Behavioral jealousy	20.777	<0.001	−1.115	−1.984, −0.247	−2.523	0.012	29.42	Yes	1.454	<0.001
QRI-F Total	Gender	MJS Behavioral jealousy	9.868	<0.001	1.324	0.103, 2.545	2.129	0.034	20.75	Female	1.791	<0.001

F *=* F distribution; *p* = *p*-value; β = standardized beta; CI = confidence interval; *t* = *t*-test.

**Table 10 behavsci-14-00044-t010:** Comparisons of MJS and QRI-F means according to gender.

	Gender	N	Mean	Standard Deviation	Mean Standard Error		Z	Sig.	*t*	df	*p*	d	d Standard Error
MJS Total	Male	176	32.14	10.81	0.82	Equal variances assumed	0.345	0.557	−1.281	660	0.200	−0.113	0.088029
	Female	486	33.38	11.13	0.50	Equal variances not assumed			−1.299	318,067	0.195		
Cognitive jealousy	Male	176	8.53	4.07	0.31	Equal variances assumed	1.772	0.184	1.675	660	0.094	0.147	0.088067
	Female	486	7.95	3.88	0.18	Equal variances not assumed			1.638	297,489	0.102		
Behavior jealousy	Male	176	7.07	2.69	0.20	Equal variances assumed	0.039	0.843	−2.332	660	0.020	−0.205	0.088154
	Female	486	7.61	2.62	0.12	Equal variances not assumed			−2.303	302,855	0.022		
Emotional jealousy	Male	176	14.19	6.45	0.49	Equal variances assumed	13.866	<0.001	−1.923	660	0.055	−0.169	0.088097
	Female	486	15.43	7.61	0.35	Equal variances not assumed			−2.077	362,878	0.039		
QRI-F Total	Male	176	2.67	0.39	0.03	Equal variances assumed	6.256	0.013	−2.585	660	0.010	−0.227	0.088195
	Female	486	2.74	0.32	0.01	Equal variances not assumed			−2.335	261,162	0.020		
Support	Male	176	3.18	0.62	0.05	Equal variances assumed	2.169	0.141	−4.942	660	<0.001	−0.435	0.088783
	Female	486	3.43	0.55	0.02	Equal variances not assumed			−4.661	279,519	<0.001		
Conflict	Male	176	2.10	0.57	0.04	Equal variances assumed	0.148	0.701	2.172	660	0.030	0.191	0.088130
	Female	486	1.99	0.56	0.03	Equal variances not assumed			2.162	307,371	0.031		
Depth	Male	176	3.11	0.62	0.05	Equal variances assumed	3.298	0.070	−4.367	660	<0.001	−0.384	0.088605
	Female	486	3.33	0.54	0.02	Equal variances not assumed			−4.097	277,119	<0.001		

*Note*: Z = standardized Levene’s test; Sig = significance; *t* = *t*-test; df = degrees of freedom; *p* = *p*-value; d = Cohen’s d size effect; M = mean; SD = standard deviation.

## Data Availability

The data presented in this study are available on request from the corresponding author.

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
