# Peer review of "Measurement Invariance of the Multidimensional Jealousy Scale and Quality of Relationships Inventory (Friend)"

_behavsci, 2024, doi:10.3390/bs14010044_

Round 1

Reviewer 1 Report

Comments and Suggestions for Authors

Major

-       I appreciate the concise introduction the authors present. However, more information is necessary for both measures that are analyzed within the article. I suggest the inclusion of one or two paragraphs informing the readers how both measures usually behave in a broad context and in different countries. This would further ensure a complete understanding of why this specific research is being performed.

-       The authors mainly focus on friendship throughout the introduction, however, their measure for jealousy is exclusively for romantic relationships. Thus, I suggest the inclusion and comparison of how both traits are displayed in different settings (i.e., friends vs. romantic). This is briefly mentioned in the limitations.

-       Regarding data analysis, the authors opted for an ML method of estimation. I highly suggest the authors review their method of estimation choice. ML assumes a continuous distribution, with previous evidence (Li, 2016) indicating its poor performance in factor models. Preferably, a WLMSV method of estimation would be the most adequate. 

Li, C. H. (2016). Confirmatory factor analysis with ordinal data: Comparing robust maximum likelihood and diagonally weighted least squares. Behavior Research Methods, 48, 936-949.

Minor

-       I strongly suggest that the authors review their title; currently, it is too long, and most of the information is already stated in the abstract. Furthermore, keywords already appearing on the title do not maximize the article’s reach once published, so I suggest new keywords.

Author Response

Dear Reviewer 1,

The authors appreciate the opportunity to review and enhance our work. We made an effort to implement all the suggestions, which we highly value. In the document highlighted in yellow, you will find our responses, and in the text, the suggested changes.

Comments and Suggestions for Authors

Major

-       I appreciate the concise introduction the authors present. However, more information is necessary for both measures that are analyzed within the article. I suggest the inclusion of one or two paragraphs informing the readers how both measures usually behave in a broad context and in different countries. This would further ensure a complete understanding of why this specific research is being performed.

This was done:

In the introduction we wrote “Croucher et al. (2012a) discovered that individuals hailing from countries that pri-oritize self-centered thinking, a trait more prevalent in cultures with a masculine orien-tation, tend to display a greater tendency towards jealousy. This increased self-centered thinking stems from a inclination to focus more on one's own well-being and less on the well-being of the couple (Croucher et al., 2012a). Notably, both Ireland and the United States generally exhibit more masculine cultural characteristics compared to India and Thailand. Consequently, it is plausible that Americans and Irish individuals might man-ifest a higher degree of jealousy, given that jealousy is viewed as an emotional manifes-tation of competition (Croucher et al., 2012a). Furthermore, Croucher et al. (2012b) con-ducted a comparative analysis of jealousy between India and the United States. Note-worthy variations emerged between men and women across all facets of jealousy. Spe-cifically, Indians reported lower levels of both cognitive and emotional jealousy in com-parison to Americans. Croucher et al. (2012b) also identified religion as a crucial factor predicting jealousy. Hindus exhibited higher levels of both cognitive and emotional jealousy, while Christians demonstrated greater cognitive jealousy than Muslims”.

And later on “According to Krappmann (1996), although there are shared cultural aspects in the understanding of friendship across various societies, distinctions exist in the interpretation and purpose of friendship based on cultural differences. Within societies and across genders, the definition of friendship varies due to societal influences. It appears that in (modern) Western societies, close friendships are often regarded as personal connections relatively unaffected by societal norms. Regarding Beer (2001), in subsistence economies, where resource distribution is not guaranteed, friendships tend to be more instrumentally oriented, focusing on material exchanges”.

And in metodology, when we presente the instruments, we wrote “This instrument has been validated for different populations, including the Urdu population (Anjum et al., 2023), Italian population (Diotaiuti et al., 2022), Iranian population (Rahimi & Sanatnama, 2021), Serbian population (Tošić-Radev & Hedrih, 2017), and Australian population (Elphinston et al., 2011)”. Later on, we wrote “Further studies conducted with samples representing diverse ethnic and cultural back-grounds have yielded consistent findings comparable to the initial validation (Marques et al., 2015; Neves & Pinheiro, 2009; Reiner et al., 2012; Verhofstadt et al., 2006). In contrast, certain investigations employing similarly diverse samples and various relationship types have revealed a two-factor solution, indicating a departure from the originally proposed factor structure of the Quality of Relationship Inventory (QRI) (Matos et al., 2013; Nakano et al., 2002)”.

-       The authors mainly focus on friendship throughout the introduction, however, their measure for jealousy is exclusively for romantic relationships. Thus, I suggest the inclusion and comparison of how both traits are displayed in different settings (i.e., friends vs. romantic). This is briefly mentioned in the limitations.

This was done. We wrote: “While research often focuses on romantic jealousy due to societal emphasis, there are shared aspects with jealousy in friendships (Worley & Samp, 2014). Exploring jealousy in friendships is essential, as emotional experiences and mechanisms are not exclusive to romantic relationships (Attridge, 2013). This broader understanding contributes to comprehensive theories of jealousy, applicable beyond romance. Recognizing this relevance informs practical interventions in counseling, education, and relationship management. In essence, acknowledging jealousy's broader significance enhances our understanding of human emotions and relationships”.

-       Regarding data analysis, the authors opted for an ML method of estimation. I highly suggest the authors review their method of estimation choice. ML assumes a continuous distribution, with previous evidence (Li, 2016) indicating its poor performance in factor models. Preferably, a WLMSV method of estimation would be the most adequate.

Li, C. H. (2016). Confirmatory factor analysis with ordinal data: Comparing robust maximum likelihood and diagonally weighted least squares. Behavior Research Methods, 48, 936-949.

This was done: In Data Analysis section we wrote: “The weighted least squared means and variance adjusted (WLSMV) method (that is designed for ordinal data according to Li, 2016). was applied for estimation”. In the Results section, revisions are highlighted in yellow, indicating changes made to both tables and corresponding text. Fortunately, the results have changed, but not the conclusions that we can draw from them.

Minor

-       I strongly suggest that the authors review their title; currently, it is too long, and most of the information is already stated in the abstract. Furthermore, keywords already appearing on the title do not maximize the article’s reach once published, so I suggest new keywords.

The authors concur and believe the following title is fitting: Measurement invariance of the Multidimensional Jealousy Scale and Quality of Relationships Inventory (Friend)

Our new keywords are: equivalence, factor, interpersonal quality, jealousy, validation

Reviewer 2 Report

Comments and Suggestions for Authors

The submitted manuscript investigates the jealousy scale and the quality of relationship inventory in a Portuguese sample. The purpose of the paper is clear, and the manuscript is well-structured. I have a few requests that should be addressed before publication.

Major comments:

1.    70: I disagree that measurement invariance (MI) is necessary for valid comparisons between groups (see Funder & Gardiner, 2023; Welzel & Inglehart, 2016). In contrast, it would be quite awkward to assume that different measurement indicators would function similarly for different ages.
2.    77: There aren’t “true differences” unless particular factor models are assumed. Hence, it is tautological to speculate about the truth because what you define as “truth” are merely particular identification constraints in the model.
3.    171: I suggest using MLR for estimating standard errors and chi-square statistics. Maybe this was already used as a default of the software?
4.    200ff.: The partial invariance technique is dubious because it is unclear why the pattern of noninvariant model parameters should follow a sparse distribution. If only a few violations of MI are postulated, invariance alignment is a much more principled and preferred method. However, there are good reasons to conduct model comparisons despite the fact that MI is violated.
5.    253ff.: It is unclear how continuous covariates of age, education, and being in a romantic relationship are treated in the MI analysis. It seems that a discretization into groups is required. The choice is arbitrary. In principle, local structural equation modeling (LSEM) and moderated nonlinear factor analysis (MNLFA) are more suitable techniques compared to multiple-group confirmatory factor analysis. At least, these techniques should be included in the discussion techniques as preferred methods.
6.    260ff.: Also, include stratified Cronbach’s alpha and omega total for the total scale that includes the three dimensions, respectively.
7.    Tables 2, 5: Please include a chi square difference test for model comparisons.
8.    Table 6: Provide standard errors for Pearson correlation. Write “Pearson correlations” instead of “Pearson’s correlations”.
9.    Table 6: Report all regression coefficients (including the nonsignificant ones). Note that the difference between significant and nonsignificant results can frequently be nonsignificant.
10.    Table 6: It is better to label “EP B” as “SE” (standard error of B).
11.    Table 10: Always perform a t-test assuming nonequal standard deviations. Report the group-specific standard deviations. Also, report a standard error for Cohen’s d.

Minor comments:

12.    Section 2: Some subsections must be numbered as “2.x.” (e.g., 2.2., and so on).
13.    213: Write “p < 0.05” instead of “p <= 0.05”.
14.    242: Avoid the line break after “17”.
15.    Throughout the manuscript and 240: Write “confirmatory factor analysis”.

Funder, D., & Gardiner, G. (2023). Misgivings About Measurement Invariance. PsyArXiv, 11 May 2023. https://osf.io/preprints/psyarxiv/97cxg
Welzel, C., & Inglehart, R. F. (2016). Misconceptions of measurement equivalence: Time for a paradigm shift. Comparative Political Studies, 49, 1068-1094

Author Response

Dear Reviewer 2:

The authors appreciate the opportunity to review and enhance our work. We made an effort to implement all the suggestions, which we highly value. In the document highlighted in yellow, you will find our responses, and in the text, the suggested changes.

The submitted manuscript investigates the jealousy scale and the quality of relationship inventory in a Portuguese sample. The purpose of the paper is clear, and the manuscript is well-structured. I have a few requests that should be addressed before publication.

Major comments:

1.70: I disagree that measurement invariance (MI) is necessary for valid comparisons between groups (see Funder & Gardiner, 2023; Welzel & Inglehart, 2016). In contrast, it would be quite awkward to assume that different measurement indicators would function similarly for different ages.

In Introduction, the following information has been added: “However, some authors disagree that measurement invariance (MI) is necessary for valid comparisons between groups (Funder & Gardiner, 2023; Welzel & Inglehart, 2016). Funder and Gardiner (2023) contend that recent discoveries challenging the widespread assumption of profound cultural differences suggest that presuming substantial cross-cultural differences in measurement should not be the default stance. Furthermore, the authors recommend a transition towards external validity as a more meaningful measure of measurement quality. As per Welzel and Inglehart (2016), constructs may not necessarily converge at the individual level but can still demonstrate significant and impactful associations at the aggregate level. Hence, the authors propose a shift in paradigm, emphasizing external linkage over internal convergence as the primary criterion for validity”

  1. 77: There aren’t “true differences” unless particular factor models are assumed. Hence, it is tautological to speculate about the truth because what you define as “truth” are merely particular identification constraints in the model.

You are right, so we removed the expression.

  1. 171: I suggest using MLR for estimating standard errors and chi-square statistics. Maybe this was already used as a default of the software?

The weighted least squared means and variance adjusted (WLSMV) method (that is designed for ordinal data according to Li, 2016) was applied for estimation (replacing ML), as suggested by reviewer 1. All the results changed but, fortunately, not the conclusions that we can draw from them. Changed results are in yellow in text.

  1. 200ff.: The partial invariance technique is dubious because it is unclear why the pattern of noninvariant model parameters should follow a sparse distribution. If only a few violations of MI are postulated, invariance alignment is a much more principled and preferred method. However, there are good reasons to conduct model comparisons despite the fact that MI is violated.

You are right, but this issue no longer applies as, by changing the results with the new estimation method, full invariance was found.

  1. 253ff.: It is unclear how continuous covariates of age, education, and being in a romantic relationship are treated in the MI analysis. It seems that a discretization into groups is required. The choice is arbitrary. In principle, local structural equation modeling (LSEM) and moderated nonlinear factor analysis (MNLFA) are more suitable techniques compared to multiple-group confirmatory factor analysis. At least, these techniques should be included in the discussion techniques as preferred methods.

Age was stratified into two groups: older and younger, with the cutoff point set at the mean value plus the standard deviation. Similarly, education was categorized into two groups: individuals with and without university studies. Concerning relationship status, this variable was inherently dichotomous, indicating whether participants were currently in a romantic relationship at the time of responding to the questionnaire. In Data Analysis, this information was added: “The variables employed for assessing measurement invariance underwent categorization: Age was divided into two groups—older and younger—with the cutoff point established at the mean value plus the standard deviation. Likewise, education was segregated into two categories: those with and without university studies. As for relationship status, this variable inherently operated in a dichotomous manner, signifying whether participants were presently engaged in a romantic relationship at the time of questionnaire completion”. Besides, in Discussion, this information was added: “However, some authors consider that the selection of the measurement invariance method appears to lack a clear rationale. In theory, local structural equation modeling (LSEM) (Robitzsch , 2023) and moderated nonlinear factor analysis (MNLFA) (Bauer, 2017; Kolbe et al., 2022) are considered more appropriate methodologies when compared to the use of multiple-group confirmatory factor analysis.”

  1. 260ff.: Also, include stratified Cronbach’s alpha and omega total for the total scale that includes the three dimensions, respectively.

Although I had tried to calculate the stratified alpha and omega, I couldn't do it. I am a psychologist, not a statistician; I have never been asked to do such a thing. From what I understand, it can only be calculated in R, and I am not proficient in R. I am sorry.

  1. Tables 2, 5: Please include a chi square difference test for model comparisons.

This was done in Table 2 and Table 5.

  1. Table 6: Provide standard errors for Pearson correlation. Write “Pearson correlations” instead of “Pearson’s correlations”.

This was done throughout the text.

  1. Table 6: Report all regression coefficients (including the nonsignificant ones). Note that the difference between significant and nonsignificant results can frequently be nonsignificant.

This was done:

Table 8. Sociodemografic and Quality of Relationships Inventory (Friend) variables that contribute to explain Multidimensional Jealousy Scale

MJS Total

Cognitive jealousy

Behavior jealousy

Emotional jealousy

B

SE

β

B

SE

β

B

SE

β

B

SE

β

Gender

1.834

0.965

0.073

-0.238

0.337

-0.027

0.603

0.231

0.101

1.265

0.661

0.076

Age

0.015

0.054

0.011

-0.042

0.019

-0.088

0.020

0.013

0.062

0.041

0.037

0.046

Education

0.323

0.276

0.045

0.006

0.096

0.002

-0.056

0.066

-0.033

0.392

0.189

0.082

Romantic relationship

-0.596

0.998

-0.023

0.409

0.349

0.045

-0.352

0.239

-0.057

-0.899

0.684

-0.053

QRI Total

Support

0.506

1.160

0.026

0.287

0.405

0.042

0.209

0.278

0.045

-0.009

0.795

-0.001

Conflict

5.735

0.780

0.292

2.339

0.273

0.334

1.325

0.187

0.281

1.280

0.535

0.098

Depth

-0.645

1.110

-0.034

-0.621

0.388

-0.091

-0.055

0.266

-0.012

0.252

0.760

0.020

R2 (R2 Adj.)

0.086 (0.083)

0.117 (0.114)

0.084 (0.081)

0.025 (0.021)

F for change in R2

60.367**

82.926**

54.338**

6.862*

R2 = R squared; R2 Adj. = R squared adjusted; B = unstandardized regression coeficients; EP B = unstandardized error of B; β = standardized regression coefficients; *p < 0.010; **p < 0.001; excluded from de model

  1. Table 6: It is better to label “EP B” as “SE” (standard error of B).

This was done.

  1. Table 10: Always perform a t-test assuming nonequal standard deviations. Report the group-specific standard deviations. Also, report a standard error for Cohen’s d.

This was done in table 10.

Table 10. Comparisons of MJS and QRI-F means according to gender

Gender

N

Mean

Standard

deviation

Mean standard error

Z

Sig.

t

df

p

d

d Standard error

MJS Total

Male

176

32.14

10.81

0.82

Equal variances assumed

0.345

0.557

-1.281

660

0.200

-0.113

0.088029

Female

486

33.38

11.13

0.50

Equal variances not assumed

-1.299

318,067

0.195

Cognitive jealousy

Male

176

8.53

4.07

0.31

Equal variances assumed

1.772

0.184

1.675

660

0.094

0.147

0.088067

Female

486

7.95

3.88

0.18

Equal variances not assumed

1.638

297,489

0.102

Behaviour jealousy

Male

176

7.07

2.69

0.20

Equal variances assumed

0.039

0.843

-2.332

660

0.020

-0.205

0.088154

Female

486

7.61

2.62

0.12

Equal variances not assumed

-2.303

302,855

0.022

Emotional jealousy

Male

176

14.19

6.45

0.49

Equal variances assumed

13.866

<.001

-1.923

660

0.055

-0.169

0.088097

Female

486

15.43

7.61

0.35

Equal variances not assumed

-2.077

362,878

0.039

QRI - F Total

Male

176

2.67

0.39

0.03

Equal variances assumed

6.256

0.013

-2.585

660

0.010

-0.227

0.088195

Female

486

2.74

0.32

0.01

Equal variances not assumed

-2.335

261,162

0.020

Support

Male

176

3.18

0.62

0.05

Equal variances assumed

2.169

0.141

-4.942

660

<0.001

-0.435

0.088783

Female

486

3.43

0.55

0.02

Equal variances not assumed

-4.661

279,519

<0.001

Conflict

Male

176

2.10

0.57

0.04

Equal variances assumed

0.148

0.701

2.172

660

0.030

0.191

0.088130

Female

486

1.99

0.56

0.03

Equal variances not assumed

2.162

307,371

0.031

Depth

Male

176

3.11

0.62

0.05

Equal variances assumed

3.298

0.070

-4.367

660

<0.001

-0.384

0.088605

Female

486

3.33

0.54

0.02

Equal variances not assumed

-4.097

277,119

<0.001

Note: Z = standardized Levene’s test; Sig = significance; t = t-test; df= degrees of freedom; p = p-value; d = Cohen´s d size effect; M = mean; SD = standard deviation

Minor comments:

  1. Section 2: Some subsections must be numbered as “2.x.” (e.g., 2.2., and so on).

This was done.

  1. 213: Write “p < 0.05” instead of “p <= 0.05”.

Sorry, but I did not find it.

  1. 242: Avoid the line break after “17”.

Sorry, the paper suffered so many changes that I do not find it.

  1. Throughout the manuscript and 240: Write “confirmatory factor analysis”.

This was done.

Funder, D., & Gardiner, G. (2023). Misgivings About Measurement Invariance. PsyArXiv, 11 May 2023. https://osf.io/preprints/psyarxiv/97cxg

Welzel, C., & Inglehart, R. F. (2016). Misconceptions of measurement equivalence: Time for a paradigm shift. Comparative Political Studies, 49, 1068-1094

Round 2

Reviewer 1 Report

Comments and Suggestions for Authors

The manuscript reads much better. However, I still have one concern regarding this previous suggestion:

-       Regarding data analysis, the authors opted for an ML method of estimation. I highly suggest the authors review their method of estimation choice. ML assumes a continuous distribution, with previous evidence (Li, 2016) indicating its poor performance in factor models. Preferably, a WLMSV method of estimation would be the most adequate.

Li, C. H. (2016). Confirmatory factor analysis with ordinal data: Comparing robust maximum likelihood and diagonally weighted least squares. Behavior Research Methods, 48, 936-949.

This was done: In Data Analysis section we wrote: “The weighted least squared means and variance adjusted (WLSMV) method (that is designed for ordinal data according to Li, 2016). was applied for estimation”. In the Results section, revisions are highlighted in yellow, indicating changes made to both tables and corresponding text. Fortunately, the results have changed, but not the conclusions that we can draw from them.

I understand that the authors updated their data analysis/results. However, if I am correct, AMOS does not offer the WLSMV estimator. Thus, I suggest the author review that section in the manuscript and present the adequate estimator. Please remember that this could alter the results if the incorrect estimator was used.

Author Response

Dear reviewer,

We employed the AMOS software for conducting confirmatory factor analysis using diagonally weighted least squares, commonly known as the asymptotically distribution-free estimator (Browne, 1984; Li, 2016). The outcomes exhibit minimal deviations from those obtained through the maximum likelihood (ML) method. Additionally, according to Li (2016), WLSMV was found to be less biased and more accurate than MLR in estimating factor loadings across various conditions. However, WLSMV showed a tendency to moderately overestimate interfactor correlations when dealing with small sample sizes or/and moderately nonnormal latent distributions. Concerning standard error estimates of factor loadings and interfactor correlations, MLR demonstrated superior performance compared to WLSMV in scenarios where latent distributions were nonnormal, especially with a small sample size of N = 200. Lastly, the proposed model tended to face over-rejection based on chi-square test statistics under both MLR and WLSMV conditions in cases of a small sample size N = 200.

As we are not proficient in the R or MPlus programs, in the event that you deem additional analyses necessary, we inquire about the feasibility of utilizing the FACTOR program (Lorenzo-Seva) or Bayesian analysis.

Browne, M. W. (1984). Asymptotically distribution-free methods for the analysis of covariance structures. British Journal of Mathematics and Statistical Psychology, 37, 62–83.

Li, C. H. (2016). Confirmatory factor analysis with ordinal data: Comparing robust maximum likelihood and diagonally weighted least squares. Behavior research methods, 48, 936-949.

Reviewer 2 Report

Comments and Suggestions for Authors

no further comments

Author Response

Despite Reviewer 2 providing a brief comment of "no further comments," the grid above indicates concerns about the research design, questions, hypotheses, methods, and presentation of results. It is unclear whether these remarks pertain to the initial review or the current one. Nevertheless, as Reviewer 1 has reiterated the need for adjustments to the estimator, the modifications we plan to implement will address Reviewer 2's recommendations eventually.